# Interaction between Rumen Epithelial miRNAs-Microbiota-Metabolites in Response to Cold-Season Nutritional Stress in Tibetan Sheep

**DOI:** 10.3390/ijms241914489

**Published:** 2023-09-23

**Authors:** Weibing Lv, Yuzhu Sha, Xiu Liu, Yanyu He, Jiang Hu, Jiqing Wang, Shaobin Li, Xinyu Guo, Pengyang Shao, Fangfang Zhao, Mingna Li

**Affiliations:** 1College of Animal Science and Technology/Gansu Key Laboratory of Herbivorous Animal Biotechnology, Gansu Agricultural University, Lanzhou 730070, China; lwb18794850948@163.com (W.L.); shayz@st.gsau.edu.cn (Y.S.); huj@gsau.edu.cn (J.H.); wangjq@gsau.edu.cn (J.W.); lisb@gsau.edu.cn (S.L.); guoxy@st.gsau.edu.cn (X.G.); shaopy@st.gsau.edu.cn (P.S.); zhaofangfang@gsau.edu.cn (F.Z.); limn@gsau.edu.cn (M.L.); 2College of Animal and Veterinary Sciences, Southwest Minzu University, Chengdu 610041, China; 3School of Fundamental Sciences, Massey University, Palmerston North 4410, New Zealand; y.h@massey.ac.nz

**Keywords:** Tibetan sheep, miRNAs, rumen epithelium, rumen microbiota, metabolites, nutritional stress

## Abstract

Tibetan sheep are already well adapted to cold season nutrient stress on the Tibetan Plateau. Rumen, an important nutrient for metabolism and as an absorption organ in ruminants, plays a vital role in the cold stress adaptations of Tibetan sheep. Ruminal microbiota also plays an indispensable role in rumen function. In this study, combined multiomics data were utilized to comprehensively analyze the interaction mechanism between rumen epithelial miRNAs and microbiota and their metabolites in Tibetan sheep under nutrient stress in the cold season. A total of 949 miRNAs were identified in the rumen epithelium of both cold and warm seasons. A total of 62 differentially expressed (DE) miRNAs were screened using FC > 1.5 and *p* value < 0.01, and a total of 20,206 targeted genes were predicted by DE miRNAs. KEGG enrichment analysis revealed that DE miRNA-targeted genes were mainly enriched in axon guidance(ko04360), tight junction(ko04530), inflammatory mediator regulation of TRP channels(ko04750) and metabolism-related pathways. Correlation analysis revealed that rumen microbiota, rumen VFAs and DE miRNAs were all correlated. Further study revealed that the targeted genes of cold and warm season rumen epithelial DE miRNAs were coenriched with differential metabolites of microbiota in glycerophospholipid metabolism (ko00564), apoptosis (ko04210), inflammatory mediator regulation of TRP channels (ko04750), small cell lung cancer (ko05222), and choline metabolism in cancer (ko05231) pathways. There are several interactions between Tibetan sheep rumen epithelial miRNAs, rumen microbiota, and microbial metabolites, mainly through maintaining rumen epithelial barrier function and host homeostasis of choline and cholesterol, improving host immunity, and promoting energy metabolism pathways, thus enabling Tibetan sheep to effectively respond to cold season nutrient stress. The results also suggest that rumen microbiota have coevolved with their hosts to improve the adaptive capacity of Tibetan sheep to cold season nutrient stress, providing a new perspective for the study of cold season nutritional stress adaptation in Tibetan sheep.

## 1. Introduction

Tibetan sheep (*Ovis aries*) are a unique sheep breed to the Tibetan Plateau that play an important role in the balance of the alpine grassland ecosystem and to the sustainable development of the local economy. Traditionally, Tibetan sheep are raised and managed by grazing, relying solely on natural pasture for their nutrient intake. The supply of forage grasses on the Tibetan Plateau is seasonally unbalanced due to various climatic conditions such as cold temperatures, low oxygen levels and strong ultraviolet rays. During the prolonged cold season (October–April of the next year), Tibetan sheep suffer from severe nutritional stress where the nutrient intake from forage falls significantly below the body’s requirements. This poses a great challenge to the survival and reproduction of Tibetan sheep [1,2]. Nevertheless, Tibetan sheep not only manage to survive and reproduce in the face of such nutritional stress in the long cold season, but also provide valuable resources, such as meat, milk, wool and other resources to local herders, which is a result of their strong cold season nutritional stress adaptation [3].

The rumen is the most important organ for nutritional digestion and absorption in ruminants, performing nutrient absorption and metabolic functions while also playing important physiological roles such as protecting the host from toxic or pathogenic factors [4]. Rumen microbiota, considered the host’s second genome can convert plant polysaccharides (starch, xylan and cellulose) into carbohydrates and proteins required by animals [5,6] and influence host nutrient and energy metabolism by facilitating food energy uptake and promoting energy storage [7,8], resulting in higher food utilization and energy transfer efficiency in ruminants [9]. In addition, rumen microbiota can modulate changes in rumen structure and physiological properties, while fermentation products of rumen microbiota have also been found to be critical for rumen papilla development [10]. Lin et al. showed that microbiota-driven production of acetate and butyric acid mediated the regulation of growth-related signaling pathways by growth-related genes in rumen epithelial cells and that the rumen microbiota and rumen hosts jointly regulated physiological processes such as rumen epithelial papilla morphology and epithelial growth [11]. Recent reports suggest that microbiota play an important role in mediating the microbiota–gut–brain signaling pathways [12].

Due to advancements in high-throughput sequencing technology, transcriptome sequencing technology (RNA-seq) has become an important tool for molecular studies of livestock growth and development, environmental adaptation, immune interactions, and molecular marker development [13,14]. miRNAs are a class of small endogenous noncoding RNAs with an average length of approximately 22 nt that are widely found in eukaryotic cells [15,16]. Mature miRNAs act as important regulators of genes and silence targeted genes mainly through direct cleavage of targeted mRNAs or inhibition of translation [17]. miRNAs have been widely demonstrated to be involved in regulating gene expression at multiple levels, including at the transcriptional, posttranscriptional, and epigenetic levels [18,19], which in turn targeted signaling proteins, nodal proteins, hormones and their receptor proteins to regulate a variety of physiological processes [15,20,21], such as cell proliferation, apoptosis, cell differentiation, lipid metabolism, and hormone secretion [22,23,24,25]. It has been reported that inhibition of the miR-30 family in intestinal epithelial cells upregulates the expression of *SOX9* mRNA, which in turn regulates the proliferation and differentiation of intestinal epithelial cells [26]. Moreover, miR-7 affects the adhesion of epithelial cells to the extracellular matrix by regulating the expression level of the transmembrane glycoprotein CD98 [27] and miR-29b induces posttranslational inhibition of low-density lipoprotein receptor-related protein 6 (LRP6) and Hu-antigen R (HuR), which play an important role in regulating intestinal homeostasis and intestinal epithelial cell (IEC) proliferation [28]. An increasing number of studies have shown that there is a tight regulatory interaction between host miRNAs and gut microbiota, that miRNA expression is dependent on gut microbiota structure [29] and that microbiota play an important role in driving the transcriptional regulation of miRNAs [30]. MiRNAs secreted into the gut by host intestinal epithelial cells, Paneth cells, and others can regulate the structure of the intestinal microbiota [31].

With the rapid development of high-throughput sequencing technology and the continuous enrichment of research tools, more researchers are combining multi-omics technology to comprehensively study specific biological processes. Currently, most multiomics-based studies have focused on human physiological functions, the immune system, and disease mechanisms, and few studies have reported on the adaptation of Tibetan sheep under cold-season nutritional stress.

In summary, this study used the RNA-seq to screen out the rumen DE miRNAs in Tibetan sheep during the cold and warm seasons, and GO- [32] and KEGG- [33] enrichment analyses were performed for DE miRNA targeted genes. The correlation between DE miRNAs and microorganisms and metabolites was further analyzed in combination with the results of the rumen microbiome [34] and metabolome [35] of Tibetan sheep in cold and warm seasons in order to answer the question of how the interactions between rumen epithelial miRNAs, rumen microorganisms, and metabolites in Tibetan sheep promote the body of Tibetan sheep to effectively respond to severe nutrient stress in the cold season. This provides a new research idea for the co-evolution of rumen microbiota and their hosts in Tibetan sheep, and also provides an important theoretical basis for understanding the mechanism of plateau adaptation in Tibetan sheep.

## 2. Results

### 2.1. Data Quality Control and miRNA Classification Annotation

At least 13,143,598 raw reads were obtained for each sample sequenced in this study, and at least 12,449,018 clean reads (12.45 M) were obtained after quality control filtering. The Q30 (%) of all sample sequencing data reached more than 96.44%, and the GC content was in the range of 49.48% to 49.75%. At least 7,242,205 (41.52%) and 4,582,673 (36.81%) clean reads from sheep samples during the cool and warm seasons were compared to the reference genome, respectively.

### 2.2. Identification of Tibetan Sheep Rumen Epithelial miRNAs

A total of 949 miRNAs were identified in this study, of which 141 were known miRNAs and 808 were novel miRNAs. MiRNAs were mainly concentrated between 21–23 nt in length, accounting for 83.04% of the total number of miRNAs, with the distribution of known miRNAs and novel miRNAs between 21–23 nt in length. The proportions were 89.36% and 81.93%, respectively (Figure 1A). Venn diagram (Figure 1B) analysis (screening expression threshold of 0.01) revealed that 574 miRNAs were coexpressed in cold and warm season rumen epithelial tissue, while 177 and 24 miRNAs were specifically expressed in the cold and warm seasons, respectively. The base preference analysis of miRNAs revealed that the highest percentage of the first base in known miRNA (Figure 1C) and novel miRNA (Figure 1E) was A base, followed by U base, while the highest percentage of base preference in each position of known miRNA (Figure 1D) and novel miRNA (Figure 1F) was U base, followed by A base.

### 2.3. Screening of Cold and Warm Season Rumen Epithelial DE miRNAs and Validation by RT-qPCR

In this study, a total of 62 DE miRNAs were screened between the cold and warm seasons using the screening criteria of fold change (FC) > 1.5 and *p* value < 0.01 (Figure 2A). Among them, 38 DE miRNAs were upregulated in the cold season, including 6 known miRNAs such as oar-miR-150, oar-miR-106b, oar-miR-200b, oar-miR-26b, oar-miR-17-5p and oar-miR-23a. And there were 24 DE miRNAs that were downregulated in the cold season, including 2 known miRNAs like oar-miR-409-3p and oar-miR-194. To further verify the accuracy of the miRNA sequencing data, 11 DE miRNAs were randomly selected from 62 DE miRNAs, and their expression was measured by RT–qPCR. These results revealed a concordance between the RT–qPCR and sequencing data, as demonstrated by the alignment of expression patterns (Figure 2B), indicating high reliability of the sequencing results and allowing for further analysis.

### 2.4. Functional Annotation of Rumen Epithelial DE miRNA Targeted Genes

A total of 20,206 targeted genes were predicted for cold and warm season DE miRNAs, among which 9019 targeted genes were predicted for known miRNAs and 19,845 targeted genes were predicted for novel miRNAs. To further analyze the molecular functions of DE miRNAs in cold and warm season rumen epithelium, GO enrichment and KEGG pathway enrichment analyses were performed on the targeted genes of DE miRNAs using GO and KEGG databases.

#### 2.4.1. GO Functional Annotation

There are 16,919 targeted genes of cold and warm season miRNAs annotated to GOs database, mainly enriched to 6362 Biological Processes (GO-BP), 979 Cellular Components (GO-CC) and 1949 Molecular Functions (GO-MF). Further study revealed that among the targeted genes significantly upregulated by miRNAs (38) in the cold season (Figure 3A), 3327 targeted genes were enriched to GO-BP process, and their gene products were mainly involved in positive regulation of GTPase activity (GO:0043547, 132 genes, *p* = 2.26 × 10^−8^), positive regulation of glycogen biosynthetic process (GO:0045725, 9 genes, *p* = 6.18 × 10^−5^), digestive tract development (GO:0048565, 15 genes, *p* = 1.10 × 10^−3^), and other biological functions. There are 3579 targeted genes enriched in GO-CC, and their gene products function mainly in stress fiber (GO:0001725, 24 genes, *p* = 2 × 10^−4^), semaphorin receptor complex (GO:0002116, 7 genes, *p* = 2 × 10^−4^), ruffle (GO:0001726, 26 genes, *p* = 9.6 × 10^−4^) and other positions. There are 3,324 targeted genes enriched in GO-MF, and the tasks performed by the gene product molecules are mainly GTPase activator activity (GO:0005096, 72 genes, *p* = 3.01 × 10^−6^), ATP binding (GO:0005524, 370 genes, *p* = 7.91 × 10^−5^), extracellular matrix structural constituent (GO:0005201, 18 genes, *p* = 8.8 × 10^−4^), and other tasks. Among the targeted genes of the cold season downregulated miRNAs (24) (Figure 3B), 3,359 targeted genes were enriched to the GO-BP process, and their gene products were mainly involved in the negative regulation of transcription from RNA polymerase II promoter (GO:0000122, 205 genes, *p* = 2.10 × 10^−12^), positive regulation of GTPase activity (GO:0043547, 120 genes, *p* = 3.43 × 10^−5^), positive regulation of protein complex assembly (GO:0031334, 10 genes, *p* = 4.17 × 10^−4^) and other biological functions. There are 3,526 targeted genes enriched in GO-CC, and the targeted gene products are mainly in the nucleoplasm (GO:0005654, 354 genes, *p* = 6.70 × 10^−4^), transcription factor complex (GO:0005667, 51 genes, *p* = 1.01 × 10^−3^), and AMPA glutamate receptor complex (GO:0032281, 12 genes, *p* = 1.17 × 10^−3^). There are 3,214 targeted genes enriched in GO-MF, and the tasks performed by the gene product molecules are mainly ATP binding (GO:0005524, 373 genes, *p* = 2.65 × 10^−5^), sequence-specific DNA binding(GO:0043565, 93 genes, *p* = 6.76 × 10^−5^), GTPase activator activity (GO:0005096, 67 genes, *p* = 8.40 × 10^−5^), and other tasks.

#### 2.4.2. KEGG Functional Annotation

In the KEGG annotation, a total of 12,631 targeted genes were annotated with cold and warm season miRNAs. A total of 5418 targeted genes were predicted by cold and warm season DE miRNAs (see Appendix A). The 38 miRNAs upregulated in the cold season and the 24 miRNAs downregulated in the cold season predicted 4517 (see Appendix A) and 3105 (see Appendix A) targeted genes, respectively. Among them, 1102 targeted genes were shared (see Appendix A).

Further analysis revealed that among the 38 miRNA targeted genes significantly upregulated in the cold season, 1667 differential targeted genes were significantly enriched in 22 signaling pathways, including axon guidance (ko04360), inflammatory mediator regulation of TRP channels (ko04750), protein digestion and absorption (ko04974), choline metabolism in cancer (ko05231), MAPK signaling pathway (ko04010), and tight junction (ko04530) pathways (Figure 4A). Among the 24 miRNA targeted genes downregulated in the cold season, 1453 differential targeted genes were significantly enriched in 26 signaling pathways, which were mainly enriched in axon guidance (ko04360), insulin resistance (ko04931), basal cell carcinoma (ko05217), tight junction (ko04530), glycerophospholipid metabolism (ko00564), microRNAs in cancer (ko05206), and phosphatidylinositol signaling system (ko04070) in the pathways (Figure 4B).

### 2.5. Correlation Analysis of Rumen Epithelial miRNA-Microbiota-Metabolite

#### 2.5.1. Correlation Analysis of Cold and Warm Season Rumen DE miRNAs with Rumen Microbiota

The correlation analysis between selected DE miRNAs involved in cold and warm season in the rumen and the rumen microbiota at the genus level (Top20) revealed a significant correlation (Figure 5). It was observed that cold and warm season rumen epithelial DE miRNAs, such as miR-106b, miR-26b, miR-150, miR-200b, miR-17-5p and miR-23a, were significantly negatively correlated with Ruminococcaceae_NK4A214 group, *Pseudobutyrivibrio*, *Succiniclasticum*, *Butyrivibrio-2*, *uncultured_bacterium_f_Muribaculaceae* (*p* < 0.05). Conversely, these miRNAs showed a significantly positively correlated with *[Eubacterium]*_coprostanoligenes *group*, *Rikenellaceae_RC9 gut group* and *Christensenellaceae_R-7 group* (*p* < 0.05). miR-194 was significantly positively correlated with *Prevotella-1*, *Ruminococcaceae_NK4A214 group*, *Pseudobutyrivibrio*, *Succiniclasticum*, *Butyrivibrio-2*, *uncultured_bacteriumf_Muribaculaceae* (*p* < 0.05) and negatively correlated with *Prevotellaceae_UCG-003*, *Prevotellaceae_UCG-001*, *uncultured_bacterium_o_WCHB1-41*, *[Eubacterium]_coprostanoligenes group*, *Rikenellaceae_RC9 gut group*, *uncultured_bacteriumf_Bacteroidales_BS11 gut group* (*p* < 0.05).

#### 2.5.2. Correlation Analysis of Cold and Warm Season Ruminal DE miRNAs with Ruminal VFAs

Results showed (Figure 6) that there was a correlation between rumen epithelial miRNAs and rumen VFAs, in which miR-106b showed a significant positive correlation with Acetate, Acetate:Propionate (A:P) (*p* < 0.05) and a significant negative correlation with Isobutyrate, Valerate (*p* < 0.05). miR-194 showed a significant negative correlation with Acetate (*p* < 0.05) and a significant positive correlation with Propionate, Isobutyrate, Butyrate, Isovalerate (*p* < 0.05). The correlations of miR-17-5p, miR-150 and miR-23a were opposite to miR-194. miR-200b was significantly positively correlated with Acetate (*p* < 0.05) and significantly negatively correlated with Isobutyrate, Butyrate, and Isovalerate (*p* < 0.05). In addition, miR-26b showed a significant negative correlation with A:P (*p* < 0.05) and miR-194, miR-409-3p showed a significant negative correlation with A:P (*p* < 0.05).

#### 2.5.3. Correlation Analysis of the Cold and Warm Season Rumen Epithelial DE miRNA Targeted Genes with Microbiota Metabolites

Furthermore, we analyzed the targeted genes of DE miRNAs in the rumen epithelial in the cold and warm seasons. These targeted genes were compared with the differential metabolites of microbiota. The results revealed that these genes and microbiota were collectively enriched in five KEGG signaling pathways (Figure 7), namely, glycerophospholipid metabolism (ko00564), apoptosis (ko04210), inflammatory mediator regulation of TRP channels (ko04750), small cell lung cancer (ko05222), and choline metabolism in cancer (ko05231).

Among them, in the choline metabolism in cancer (ko05231) KEGG pathway, *SLC22A1* and *SLC44A1* were significantly upregulated in the cold season rumen epithelial, which transports choline and participate in the glycerophospholipid metabolism (ko00564) process, leading to a significant increase in phosphatidylcholine (Lecithin) levels during the cold season; *LCAT*, which regulates 1-Acyl-sn-glycero-3-phosphocholine levels, was significantly upregulated in the cold season. *PLD1* was significantly downregulated in the cold season, leading to a significant decrease in the levels of the downstream metabolite 1,2-diacy-sn-glycerol-3P during the cold season. Among them, glycerophospholipid metabolism (ko00564) pathway showed significant increases in the phosphatidyl-L-serine level during the cold season. *PISD* was significantly upregulated in the cold season, resulting in significant increases in the levels of downstream metabolite phosphatidyl-ethanolamine in the cold season. *PEMT* was significantly upregulated in the cold season, leading to a significant increase in the downstream metabolite phosphatidylcholine (Lecithin) during the cold season. Moreover, decreased 1,2-Diacy-sn-glycerol-3P levels in the cold season led to significant upregulation of the downstream targeted genes *RAF1*, *PIP5K* and *WASF2* during the cold season, resulting in a significant downregulation of the *mTOR* gene, which in turn led to significant downregulation of the *RPS6KB2* gene in the cold season through phosphorylation. In addition, in the apoptosis pathway, the sphingosine level was significantly increased in the cold season, leading to significant down-regulation of the *CTSF*, *CTSZ* and *CTSW* downstream cathepsin family genes during the cold season.

## 3. Discussion

In this study, we comprehensively analyzed the characteristics of the transcriptome, microbiota and metabolome of Tibetan sheep rumen during the cold and warm seasons and explored the interaction between rumen epithelial miRNAs and microbiota and their metabolites in response to nutritional stress in Tibetan sheep during the cold season. In this study, a total of 62 DE miRNAs were identified in the rumen epithelium of Tibetan sheep during the cold and warm seasons, of which, 38 miRNAs were significantly upregulated and 24 miRNAs were significantly downregulated in the cold season, indicating that there were significant differences in rumen transcripts between the cold and warm seasons, which also indicated that Tibetan sheep were strictly regulated by a large number of transcriptional events under severe nutritional stress in the cold season. Subsequent GO enrichment analysis showed that the targeted genes of cold and warm season DE miRNAs were mainly enriched in the positive regulation of GTPase activity (GO:0043547), GTPase activator activity (GO:0005096) and ATP binding (GO:0005524) among other pathways. It has been reported that GTPase is a key regulator of intracellular membrane trafficking and plays a role in various cellular processes such as gene expression, cytoskeleton dynamics, cell division, and cell adhesion [36], while miRNAs can regulate the posttranscriptional processing of GTPase-encoding mRNAs [37]. This finding indicates that cold and warm season DE miRNAs can significantly regulate GTPase activity. Tight junctions, as an important component of the intestinal epithelial barrier, function mainly by maintaining the integrity of the barrier function [38], suggesting that rumen epithelial DE miRNAs in Tibetan sheep during the cold and warm seasons mainly regulate rumen epithelial barrier function and thus maintain normal physiological functions of the rumen, which is essential for Tibetan sheep to cope with cold season nutritional stress.

Microbiota play an important role in several biological processes, such as host energy utilization, nutrient assimilation and host immune function [39], and coevolution with the host influence host adaptations [40]. Ruminal microbiota are symbiotic with the host, and the VFAs produced by rumen microbiota fermentation can provide approximately 75% of the energy for ruminants and are the main source of metabolic energy for the host [41]. The ruminal microbiota structure is influenced by external factors such as diet, species and environment [42], in addition to miRNAs originating from the host [31]. Xu et al. [43] showed that microbiota have a positive role in maintaining the integrity of intestinal barrier function and that butyric acid can reduce local inflammation and improve intestinal barrier permeability by promoting mucin synthesis and tight junctional reorganization. In this study, we found that cold and warm season Tibetan sheep rumen epithelial DE miRNAs were correlated with rumen microbiota as well as VFAs. Due to the significantly higher crude fiber content of pasture foraged by Tibetan sheep during the cold season, there is an increase in the levels of acetate in the rumen and a decrease in the abundance of *Prevotella-1* [44]. Acetate is used as an energy source through oxidation in the tricarboxylic acid cycle or during fatty acid synthesis [45], thus maintaining the energy supply of the body. In addition, *Prevotella-1* has been associated with the production of propionate [46], which is the main precursor substance for gluconeogenesis and can synthesize glucose for the energy supply of the organism [45]. The relative abundance of the *Rikenellaceae_RC9 gut group* was significantly higher in the cold season, improving the degradation of cellulosic polysaccharides of the pasture grass to help Tibetan sheep cope with nutrient deprivation in the cool season [47,48]. *Butyrivibrio-2* and *Pseudobutyrivibrio* are cellulolytic bacteria that play a key role in degrading cellulose-like materials to produce VFAs [49,50]. Therefore, the interaction between cold and warm season Tibetan sheep rumen epithelial DE miRNAs with rumen microbiota and VFAs is mainly to improve the degradation of cellulose-like substances and produce as many metabolites as possible to provide energy for the host, thus effectively responding to nutrient stress in the cold season.

The targeted genes of DE miRNAs in the rumen epithelial during the cold and warm seasons are coenriched with microbiota metabolites in five KEGG signaling pathways that jointly regulate multiple biological processes. Both *SLC22A1* and *SLC44A1* belong to the solute carrier family (SLC) [51]. The *SLC22A1* transporter protein plays an important role in endogenous metabolites, and other compound *SLC22A1* transporter proteins play an important role in the transport of endogenous metabolites and other compounds [52]. The *SLC44A1* protein is an important mediator of choline transport across the plasma membrane and mitochondrial membrane [53]. Choline is an important component in the synthesis of membrane phosphatidylcholine and the neurotransmitter acetylcholine, and the accumulation of choline in the cell is an important rate-limiting step in phospholipid metabolism [54]. In the choline metabolism in cancer (ko05231) pathway, *SLC22A1* and *SLC44A1*, located in the cell membrane transporter, were significantly upregulated in the cold season, which could significantly improve the efficiency of the transporter’s transmembrane transport of the metabolite choline and transport more choline into the cell to participate in the glycerophospholipid metabolism cycle. *PISD* is a key enzyme in the synthesis of phosphatidylethanolamine (PE) and plays a central role in the metabolism of phospholipids [55]. PEMT catalyzes the methylation of PE into phosphatidyl choline (PC), which is further converted into choline by intracellular lipid metabolism. When the body is nutrient deficient, the *PEMT* pathway is able to maintain sufficient levels of PC and choline to ensure the body’s energy metabolic needs [56]. In the present study, levels of the glycerophospholipid metabolism intermediate phosphatidylcholine (Lecithin) were significantly higher in the cold season, and PISD was significantly upregulated in the cold season, catalyzing the production of phosphatidyl-L-serine to produce more phosphatidyl-ethanolamine, which is further catalyzed by the significant upregulation of *PEMT*, resulting in high levels of phosphatidylcholine in the organism. This process not only maintains the stability of the glycerophospholipid metabolism (ko00564) pathway, but also ensures the energy supply of the organism and improves its ability to cope with severe nutritional stresses in the cold season. *RAF1*, as a signaling factor of the MAPK signaling pathway, directs receptor signals from the cell membrane to the nucleus and is a major transmitter of cell growth and cell proliferation signaling [57]. *RAF1* in an activated state can activate protein kinases MEK1 and MEK2 through phosphorylation, which further phosphorylates and activates serine-/threonine-specific protein kinases ERK1 and ERK2, thus activating the RAF1/MEK/ERK signaling pathway [58]. The activation of the MAPK signaling pathway may lead to tumor cell proliferation, differentiation, invasion and metastasis, accelerating the pathological process of tissue [58], while *RAF1* is also highly expressed in many types of tumor cells [59]. *WASF1* and *WASF2* belong to the WASF family and play an important role in cell invasion and migration, which are considered to be key steps in tumor metastasis [60]. It has been reported that *WASF1* is highly expressed in a variety of tumor cells such as prostate cancer [61], epithelial ovarian cancer [62] and blood cancer cells [63], and *WASF2* is also highly expressed in melanoma cells [64]; therefore, high expression of *WASF* family genes is considered a marker gene for tumor progression. *RPS6KB2* is a key gene in the mTOR signaling pathway that encodes the protein P70S6K, a substrate of mTOR. mTOR activates its downstream targeted protein P70S6K mainly by phosphorylation, which in turn phosphorylates the S6 ribosomal protein [65]. The mTOR-P70S6K signaling pathway senses the nutritional and energy status of cells and regulates cellular aging by fine-tuning the cellular response to DNA damage through signaling [66] and reduces oxidative DNA damage and thus other features of cellular aging and senescence [67]. In the present study, 1,2-diacy-sn-glycerol-3P was significantly downregulated in the cold season and the regulatory downstream genes *RAF1*, *WASF1* and *WASF2* were significantly upregulated in the cold season, resulting in significant downregulation of *mTOR* and *RPS6KB2*, suggesting that chronic activation of the MAPK signaling pathway occurs in Tibetan sheep during the cold season, which can lead to the proliferation of tumor cells. As a result, the rumen epithelium has a potential tendency to develop diseases when Tibetan sheep are under long-term nutritional stress, which is extremely unfavorable to the growth and development of Tibetan sheep.

*PLD1* is a key enzyme catalyzing the hydrolysis of phosphatidylcholine in lipid metabolism [68] and is mainly distributed on the Golgi membrane, lysosomal membrane and endoplasmic reticulum membrane in the periphery of the nucleus. When cells are subjected to external stimuli, *PLD1* is rapidly transferred to the cytoplasmic membrane to participate in lipid metabolism processes [69]. Phosphatidic acid (PA), is a direct product of the *PLD1* lipid. In the Akt/mTOR signaling pathway, PA can bind to the mTOR structural domain to stabilize the mTORC2 complex, which in turn regulates cellular physiological functions such as cell growth [70]. *PLD* can affect *mTOR* targets and may become a target of the mTOR pathway for antitumor therapy. As a target of the Wnt signaling pathway and a positive feedback regulator, β-catenin forms the β-catenin/TFC-4 complex by binding to TFC-4-binding sites (TBE) in the *PLD1* structure, thus activating *PLD* to promote tumor cell proliferation and invasion [71]. *LCAT* is a key enzyme in lipoprotein metabolism and is involved in the reversal of cholesterol transport and high-density lipoprotein cholesterol (HDL-C) metabolism, which is important for maintaining cholesterol homeostasis and regulating cholesterol transport [72]. *PIP5K* is able to phosphorylate phosphatidylinositol 4-phosphate (PI4P) with the assistance of Rho, ARFs and Rac proteins in the small G protein family to produce phosphatidylinositol 4,5-bisphosphate (PI (4, 5) P2) [73]. PI (4, 5) P2 is a phospholipid molecule widely distributed in cell membranes. Unoki et al. [74] found that PIP5K kinase can induce increased production of PI(4, 5)P2 upon stimulation by external pathogens, which in turn activates the Toll-like receptor 4 (TLR4) immune signaling pathway. Cathepsin, as a protein hydrolase that mainly cleaves peptide bonds in living organisms, is mainly found in lysosomes and can be involved in various important physiological activities such as protein hydrolysis, antigen presentation, cell regulation, and embryonic development [75]. While *CTSF* and *CTSZ* both belong to the cathepsin family, *CTSF* is mainly involved in the presentation of type II histocompatibility complex (synthesized by R lymphocytes and macrophages) antigens (production of corresponding antibodies to counteract heterologous proteins), thus participating in the autoimmune response of the body [76]. High *CTSW* expression can inhibit cancer cell progression and is associated with antiviral properties [77]. In the present study, the *PLD1* was significantly downregulated in the cold season, which could effectively reduce the PA content in lipid metabolism, thereby inhibiting tumor cell genesis and proliferation through the Akt/mTOR signaling pathway and Wnt signaling pathway. *LCAT* and *PIP5K* were both significantly upregulated in the cold season, which could maintain cholesterol homeostasis and promote lipoprotein metabolism, and *CTSF* and *CTSW* were significantly upregulated in the cold season, which could enhance the host’s autoimmune response and effectively inhibit the process of rumen epithelial cell carcinogenesis. Therefore, when Tibetan sheep are exposed to a long-term cold environment and severe nutritional stress, miRNAs can regulate the expression of targeted genes to enhance host autoimmunity and inhibit the process of cell carcinogenesis, thus alleviating rumen epithelial disease development and enabling Tibetan sheep to effectively cope with cold season nutritional stress in the Tibetan Plateau.

In summary, when Tibetan sheep are subjected to long-term cold-season nutrient stress, the interaction between rumen epithelial miRNAs, rumen microbiota and their metabolites and the host undergoes a multipathway signaling exchange, which together regulate a series of fine, complex and ordered physiological activities in the rumen epithelium to effectively respond to cold-season nutrient stress (Figure 8). In Tibetan sheep subjected to cold-season nutrient stress, the expression of the targeted genes *RAF1*, *mTOR* and *WASF2* is regulated by DE miRNAs in the rumen epithelium, which act on the MAPK signaling pathway, mTOR signaling pathway and the actin cytoskeleton pathway, respectively, which put the rumen epithelium at a potential risk of tumorigenesis and lesions. Moreover, the DE miRNAs regulate the expression of targeted genes *PLD1*, *PIP5K* and cathepsin family genes (*CTSF* and *CTSW*) to enhance the host autoimmunity and inhibit cellular carcinogenesis, thus alleviating the process of the rumen epithelial disease and maintaining the healthy state of rumen epithelium. Ruminal epithelial DE miRNAs regulate the expression of targeted genes *SLC22A1*, *SLC44A1*, *PISD* and *PEMT*, transport the microbiota metabolites choline to participate in intracellular glycerophospholipid metabolism, and act through the Wnt signaling pathway to maintain choline, 1,2-diacyl-sn-glycerolI-3P and cholesterol homeostasis, which not only ensures the energy supply of the body, but also maintains the homeostasis of the body microenvironment. In return, miRNAs provide more energy to the host by regulating the structure of rumen microflora, which in turn regulates the mode of fermentation in the rumen and improves the efficiency of translocation and uptake of metabolite VFAs, enabling Tibetan sheep to effectively cope with cold season nutrient stress. The above results also suggest that rumen microbiota and their hosts have coevolved to participate in the adaptation and regulation of cold season nutrient stress in Tibetan sheep, providing a new perspective for the study of cold season nutritional stress adaptation in Tibetan sheep.

## 4. Materials and Methods

### 4.1. Experimental Design and Sample Collection

Tibetan sheep are selected from the same herding pasture in Hezuo City, Gannan Tibetan Autonomous Prefecture, Gansu Province, China (3300 m above sea level). Tibetan sheep were raised and managed under traditional natural grazing conditions without any supplemental feeding. In January 2019, we selected 12 Tibetan sheep (females, not pregnant) aged 1 year (12 ± 1 month old), of similar weight (35.12 ± 1.43 kg) and in good health, and they are mixed into the flock and managed uniformly for grazing. Six Tibetan sheep were randomly selected in July (warm season) and December (cold season) 2019, respectively, and slaughtered using the traditional carotid artery bleeding method in the morning before grazing, and immediately after dissection, ruminal ventral capsule tissue (*n* = 6 in the warm season and *n* = 6 in the cold season) was taken, the contents were quickly rinsed with PBS, and then quickly placed in liquid nitrogen and brought back to the laboratory for storage at −80 °C for subsequent transcriptome sequencing and extraction of total RNA. At the same time, we collected the rumen contents (*n* = 6 in warm season and *n* = 6 in cold season) of Tibetan sheep using measuring cylinders (sterilized and sterilized), and filtered through sterile gauze and dispensed in lyophilized tubes, rapidly frozen in liquid nitrogen, and brought back to the laboratory for storage at −80 °C for total microbiota DNA extraction, microbiota 16S rRNA sequencing, and VFAs content determination.

### 4.2. Microbiota 16S rRNA Sequencing of Rumen Contents in Tibetan Sheep

The frozen rumen contents were thawed at room temperature, and an appropriate amount was taken in a 2 mL centrifuge tube and centrifuged at 1200 rpm for 2 min. The supernatant was transferred to a new 2 mL centrifuge tube, and then the total DNA in each sample was extracted according to the procedure of MN NucleoSpin 96 Soil kit (MACHEREY-NAGEL GmbH & Co. KG, Düren, Germany). Next, 16S rRNA genes in the V3–V4 region of the bacteria were amplified using primers 338F (5′-ACTCCTACGGGAGGCAGCAG-3′) and 806R (5′-GGACTACHVGGGTWTCTAAT-3′). Finally, all amplification products were sequenced and analyzed using the Illumina MiSeq platform (Illumina, San Diego, CA, USA). In this study, the genus level (the 20 species with the highest relative abundance) of microbiota 16S rRNA sequencing results were selected for correlation analysis.

### 4.3. Determination of VFAs in Rumen Contents of Tibetan Sheep

The internal standard method was adopted, using 2-ethyl butyric acid (2EB) as the internal standard. Rumen contents frozen at −80 °C were thawed and centrifuged at room temperature. 1 mL of supernatant was pipetted into a 1.5 mL centrifuge tube and placed on ice to await analysis. Then, 0.2 mL of 25% metaphosphoric acid solution containing internal standard 2EB was added. Then, mix, ice bath for 30 min, centrifuge at 10,000 rpm for 10 min, filter the supernatant into a sample bottle with a 0.25 μm filter, and the amount of VFAs determined by the gas chromatograph (GC-2010 plus; Shimadzu, Tokyo, Japan). In this study, the raw data from the VFAs measurement results were selected for correlation analysis.

### 4.4. MiRNA Sequencing of Tibetan Sheep Rumen Epithelium

#### 4.4.1. Extraction of Total RNA from Rumen Epithelial Tissue of Tibetan Sheep

Tibetan sheep ruminal epithelial tissue samples collected in the cold and warm seasons were found in the −80 °C sample bank, and three samples were randomly selected for transcriptome sequencing in the cold and warm seasons (n = 3 in the warm season and n = 3 in the cold season), respectively. Total RNA was extracted from cold and warm season Tibetan sheep rumen epithelial tissue using the Trizol reagent, and the concentration of total RNA was detected using NanoDrop 2000 (Thermo Scientific, Waltham, MA, USA) and its integrity was checked using Agilent 2100 (Model Platinum Elmer LabChip GX, Agilent, CA, USA).

#### 4.4.2. Preparation and Quantification of miRNA Libraries

To construct cold-season (n = 3) and warm-season (n = 3) Tibetan sheep rumen epithelial miRNA libraries, 1 μg of total RNA from each sample with satisfactory quality testing (concentration ≥ 200 ng/μL, OD260/280 between 1.7–2.5, OD260/230 between 0.5–2.5, RIN value ≥ 8) was selected as the starting RNA amount for library construction. The library construction kit (VAHTSTM Small RNA Library Prep Kit for Illumina, NR801-02, Vazyme, Nanjing, China) was then used to attach a universal splice to the 3′ and 5′ ends of small RNA. The first strand cDNA was then synthesized by reverse transcription, and the product was purified by PCR amplification using VAHTSTM DNA Clean Beads (N411-03), followed by polyacrylamide gel electrophoresis (PAGE), gel cutting and gel recovery to finally obtain the cold and warm season Tibetan sheep rumen epithelial miRNAs. A sheep rumen epithelial miRNA library was obtained. The sequencing process was completed by Biomarker Technologies Corporation (Beijing, China) using Illumina HiSeq3500, and raw sequenced sequences (raw reads) were obtained.

#### 4.4.3. Identification of miRNAs and Targeted Gene Prediction, Functional Annotation and Pathway Enrichment Analysis in the Rumen Epithelium of Tibetan Sheep

The raw reads were quality controlled (removing a series of noncompliant sequences such as sequences containing connectors and low-quality sequences) to obtain high-quality sequences (clean reads). The clean reads were compared with the reference genome (Ovis_aries.Oar_rambouillet_v1.0) using Bowtie (v1.0.0) software to obtain the position information (mapped reads) on the reference genome. The reads matched to the reference genome were compared with the mature sequences of known miRNAs and their upstream 2 nt and downstream 5 nt ranges in the miRBase (v22) database to identify known miRNAs. Combined with the biological characteristics of miRNAs, for sequences not identified to known miRNAs, the prediction of new miRNAs was performed using miRDeep2 (v2.0.5) software.

The set of DE miRNAs of the cold and warm season ruminal epithelium was screened (FC ≥ 1.5; *p* value ≤ 0.01) using DESeq2 (v1.6.3) software. Based on the gene sequence information of known miRNAs and novel miRNAs with corresponding species, targeted gene prediction of DE miRNAs was performed with miRanda (v3.3a) and TargetScan (v5.0). The main principles of miRNA targeted gene prediction are complementarity of seed sequences, sequence conservativeness, site binding ability, UTR base distribution, correlation between miRNA and targeted gene tissue distribution. The predicted miRNA targeted genes were compared with mRNA sequencing data, and differentially expressed genes were screened using FC ≥ 2 and FDR < 0.01 as criteria. Using the expression of the miRNA targeted gene in the warm season as a control, the rise and fall in the expression of this gene in the cold season were defined as upregulation and downregulation in the cold season, respectively (*p* < 0.05 was the screening criterion). The mRNA sequencing process and data analysis methods were reported in a previous study [35]. Then, functional annotation as well as pathway enrichment analysis of targeted genes of the cold and warm season DE miRNAs were performed using GO (http://www.geneontology.org/, accessed on 18 September 2020) [32] and KEGG (http://www.genome.jp/kegg/pathway.html, accessed on 18 September 2020) [33] databases.

### 4.5. Real-Time Quantitative PCR (RT-qPCR) Validation

Eleven DE miRNAs were randomly selected and specific primers were designed by Primer 5.0 according to the principle of additive tailing method (Table 1). The expression of DE miRNAs was detected by RT-qPCR method, and the relative expression of miRNAs was calculated by 2^−ΔΔCt^ method, and the expression of miRNAs was corrected by using U6 and 18S RNA as internal reference genes.

### 4.6. Data Analysis

Statistical analysis of RT-qPCR data was performed using the independent samples *t*-test in the IBM SPSS Statistics (V.25) software module and was statistically significant at the *p* < 0.05 level. In this study, the correlation analysis between rumen VFAs, rumen microbiota 16S rRNA and rumen epithelial miRNAs was performed by Spearman’s method: firstly, the monotonic relationship between the variable data was detected using the IBM SPSS Statistics (V.25) software; secondly, Spearman’s method was used to calculate the correlation coefficient; and finally, the two-tailed method was used to test the significance, correlation coefficient threshold 0.8, significance *p* < 0.01. The methods and results of the cold- and warm-season Tibetan sheep rumen VFAs and microbiota 16S rRNA [32] and microbiota metabolite [33] were determined and described in detail in a previous report.

## 5. Conclusions

There is an important relationship between rumen epithelial DE miRNAs and microbiota and their metabolites in Tibetan sheep, and together they mainly maintain rumen the epithelial barrier function, maintain host homeostasis of choline and cholesterol, improve host immunity, and promote energy metabolism pathways, thus enabling Tibetan sheep to effectively respond to cold-season nutritional stress. This study also suggests that the rumen microbiota and Tibetan sheep have coevolved to adapt and regulate cold-season nutrient stress, providing a new perspective on the adaptation of Tibetan sheep to cold-season nutritional stress.

## Figures and Tables

**Figure 1 ijms-24-14489-f001:**
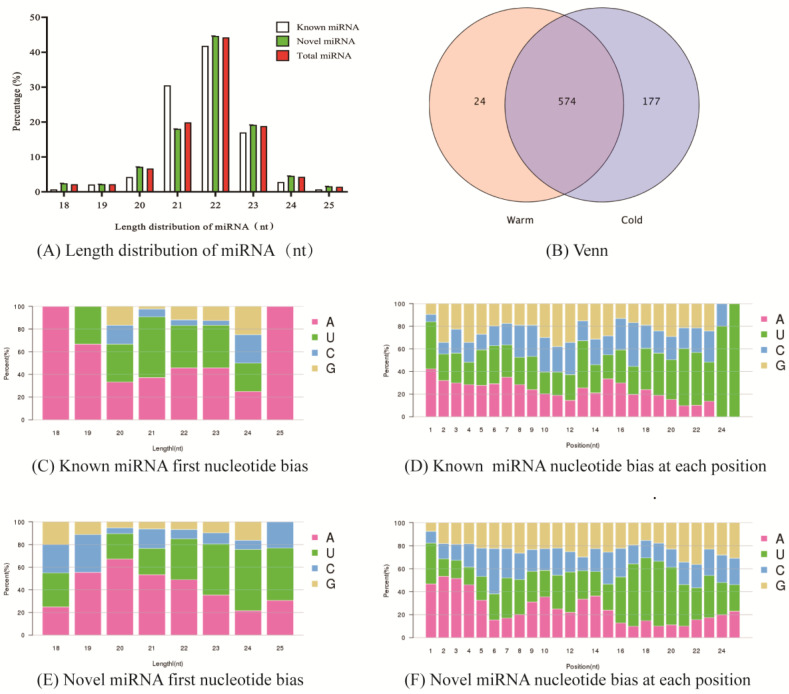
Basic characterization of miRNA.

**Figure 2 ijms-24-14489-f002:**
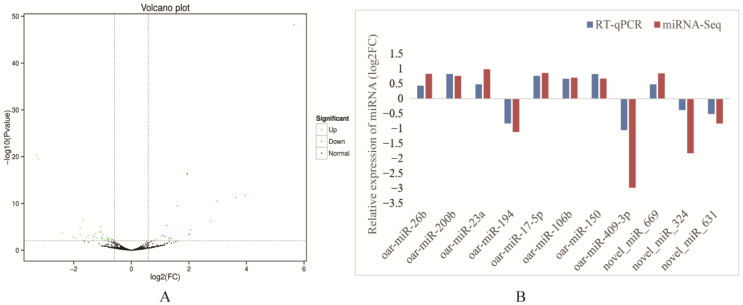
(**A**) DE miRNAs volcano plot (Up: Significant upregulation of miRNAs in the cold season; Down: Significant downregulation of miRNAs in the cold season). (**B**) RT-qPCR validation of differentially expressed miRNAs.

**Figure 3 ijms-24-14489-f003:**
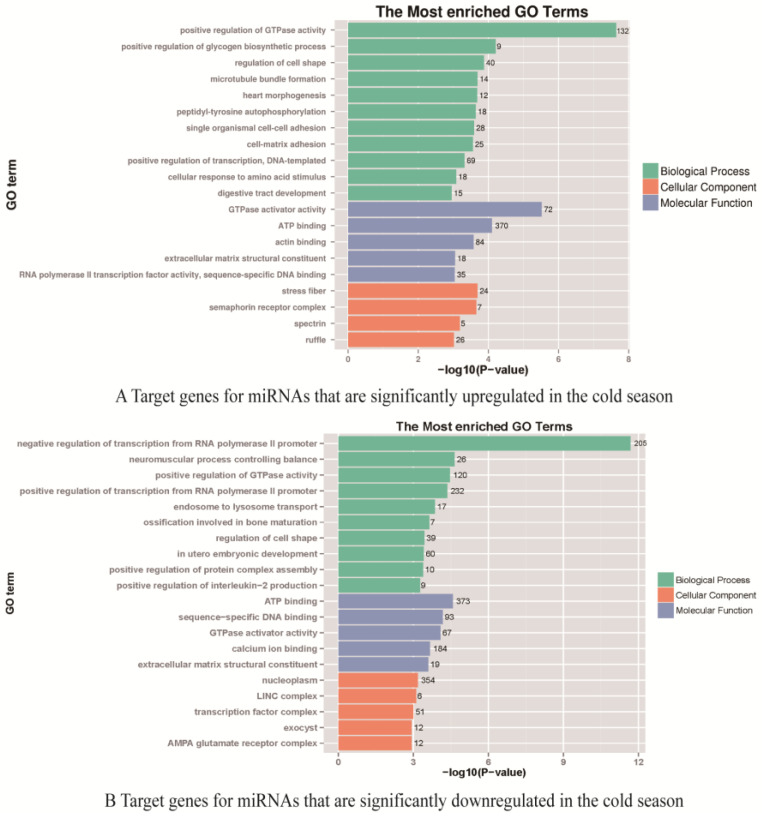
GO enrichment analysis. (**A**) Targeted genes for miRNAs that are significantly upregulated in the cold season. (**B**) Targeted genes for miRNAs that are significantly downregulated in the cold season.

**Figure 4 ijms-24-14489-f004:**
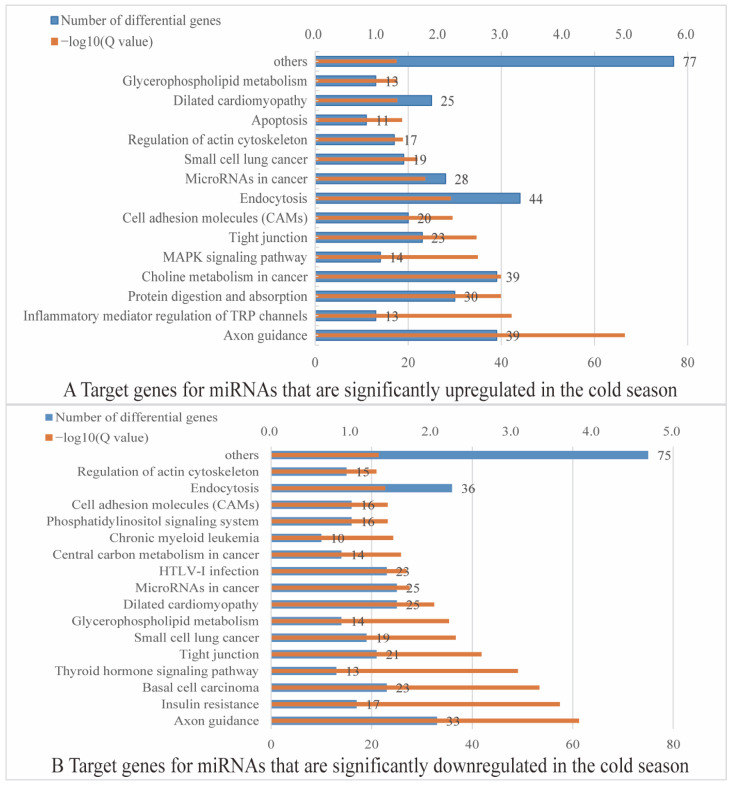
KEGG enrichment analysis. (**A**) Targeted genes for miRNAs that are significantly upregulated in the cold season. (**B**) Targeted genes for miRNAs that are significantly downregulated in the cold season.

**Figure 5 ijms-24-14489-f005:**
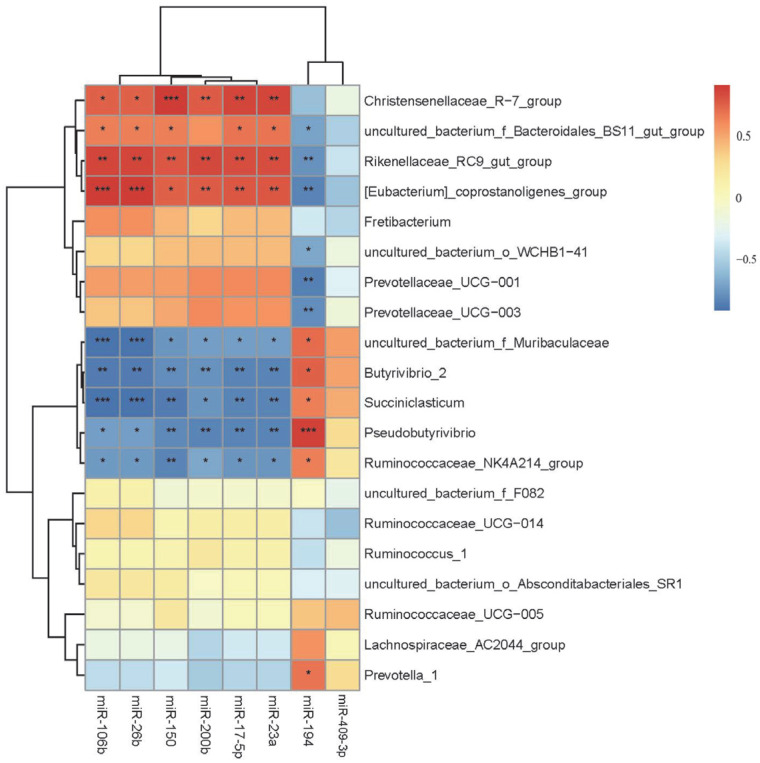
Correlation analysis between cold and warm season rumen epithelial DE miRNAs and microbiota. * Correlations differ at the 0.05 level; ** correlations differ at the 0.01 level. *** correlations differ at the 0.001 level. Same as below.

**Figure 6 ijms-24-14489-f006:**
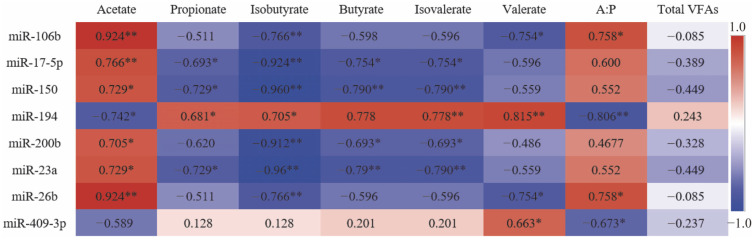
Correlation analysis between cold and warm season rumen DE miRNAs and rumen VFAs. Note: The numbers in the figure represent the correlation coefficients between DE miRNAs and rumen VFAs. Red represents a positive correlation. Blue represents a negative correlation. The darker the color, the higher the correlation. * represents significant correlation between DE miRNAs and rumen VFAs; ** represents significant correlation between DE miRNAs and rumen VFAs.

**Figure 7 ijms-24-14489-f007:**
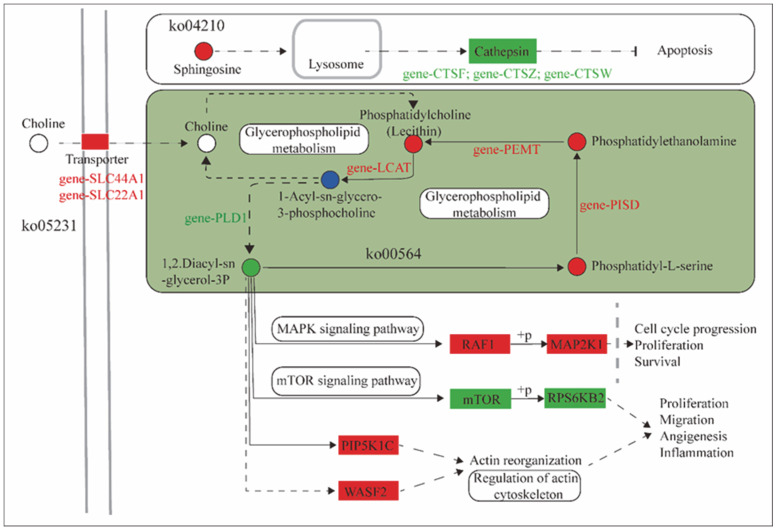
Correlation analysis between cold and warm season rumen epithelial DE miRNAs targeted genes and microbiota metabolites. Note: Red: cold upregulation. Green: cold downregulation. Blue: both upward and downward adjustments in the cold season. Circles: metabolites. Boxes: DE miRNA targeted genes. Rounded rectangles: signaling pathways. The direction of the arrow indicates the direction of the mode of action. The straight line represents direct action. The dotted line represents indirect action.

**Figure 8 ijms-24-14489-f008:**
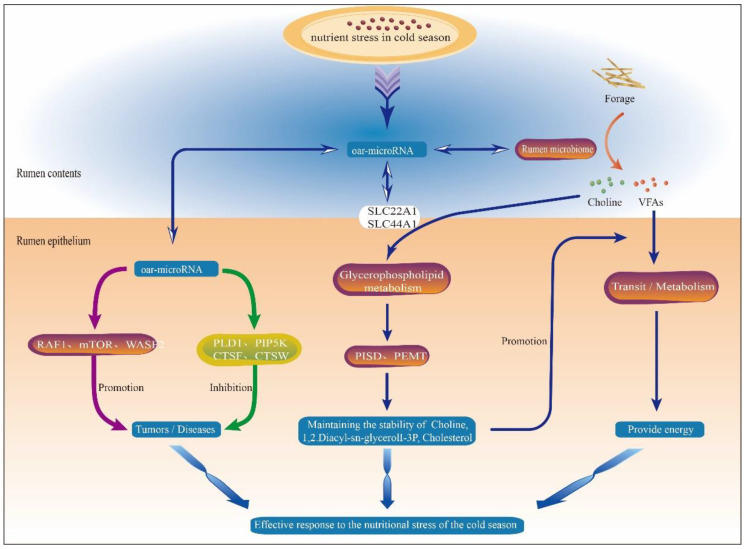
The regulatory model of DE miRNAs in response to nutrient stress in the cold season. Note: The light color in the picture (**upper part**) indicates the space where the rumen contents exist. The dark color in the picture (**lower part**) indicates the rumen epithelial tissue. The arrow represents the direction of action. Red arrows represent promotion. Green arrows represent inhibition.

**Table 1 ijms-24-14489-t001:** The primer information for RT-qPCR verification of differentially expressed miRNAs.

miRNAs/Genes	Forward Primer (5′→3′)	Reverse Primer (5′→3′)
miR-26b	TTCAAGTAATTCAGGATAGGT	Universal reverse prime r *
miR-200b	TAATACTGCCTGGTAATGATG	Universal reverse primer *
miR-23a	ATCACATTGCCAGGGATTTCCA	Universal reverse primer *
miR-194	TGTAACAGCAACTCCATGTGGA	Universal reverse primer *
miR-17-5p	CAAAGTGCTTACAGTGCAGGTA	Universal reverse primer *
miR-106b	TAAAGTGCTGACAGTGCAGAT	Universal reverse primer *
miR-150	TCTCCCAACCCTTGTACCAGTG	Universal reverse primer *
miR-409-3p	CGAATGTTGCTCGGTGAACCCCT	Universal reverse primer *
miR-669	UCCUUCAUUCCACCGGAGUCUGU	Universal reverse primer *
miR-324	GUCCAGUUUUCCCAGGAAUCCCU	Universal reverse primer *
miR-631	CUGACCUAUGAAUUGACAGCCAG	Universal reverse primer *
U6	ACGGACAGGATTGACAGATT	TCGCTCCACCAACTAAGAA
18S RNA	GTGGTGTTGAGGAAAGCAGACA	TGATCACACGTTCCACCTCATC

Note: * The universal reverse primer used for RT-qPCR amplification of miRNAs was provided by the kit (Mir-XTM mi RNA First-Strand Synthesis Kit, Takara, CA 94043, USA).

## Data Availability

The datasets presented in this study can be found in online repositories (https://www.ncbi.nlm.nih.gov/sra, accessed on 12 September 2022). The names of the repository/repositories and accession numbers can be found below: [Sequence Read Archive (SRA): PRJNA879425].

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
