# Peer review of "Interaction between Rumen Epithelial miRNAs-Microbiota-Metabolites in Response to Cold-Season Nutritional Stress in Tibetan Sheep"

_ijms, 2023, doi:10.3390/ijms241914489_

Round 1
Reviewer 1 Report
Dear Dr. Weibing,
I hope this message finds you well.
I would want to express my gratitude for the work you put into the study that was presented in the manuscript that you submitted to IJMS Journal. I understand the importance of your work in addressing the Interaction between Rumen Epithelial miRNAs-Microbiota-Metabolites in Response to Cold-Season Nutritional Stress in Tibetan Sheep.
I applaud you for your thorough study and straightforward presentation of the research findings. Your work reveals a thorough knowledge of the subject.
I encourage you to address the aforementioned points. Your work has the potential to significantly advance the field with the appropriate adjustments.
I want to thank you once more for your useful input and I'm looking forward to reading your feedback.
Sincerely,
Reviewer 2 Report
The manuscript entitled "Interaction between Rumen Epithelial miRNAs-Microbiota-Metabolites in Response to Cold-Season Nutritional Stress in Tibetan Sheep" presents a research paper about rumen epithelial miRNA DE, microbiota and its metabolites in tibetan sheep.
In my opinion this manuscript is of major scientific relevance to a better understanding of adaption mechanisms that some autochthonous sheep breeds have. We do also consider this paper is well written although extremely extensive. This topic was systematically studied which led to a paper of difficult reading. Summarizing and because of high scientific relevance and also because of important data obtained, we do consider that this paper should be accepted for publication. Also present few suggestions of editing information: - In my opinion a paragraph about clear objectives should be written and presented; - Graphics and diagrams are presented together as figures. I think this should be reviewed, because as they are presented they are not so easy to read. These should be lighter leading to an easier reading.First paragraphs should go through minor English revision, If you allow me, I'm sending few examples to improve it.
Line 13 - delete "the" rumen is an important...
Line 13 - importante nutrient FOR metabolism and AS AN absorption organ Line 18 - delete "the" cold and warm... Line 24 - the targeted metabolites Line 31 - delete "the" results... line 39 - ... ecosystem and to the sustainable development... of local economy Line 66 - the rapid development Lime 135 - these results showed... Line 209 - rewrite group names... delete underscore - not professional as presenting results - are these names in underscore - are these official bacteria names? Line 226 - delete "the" results showed... Line 241 - analysis of THE cold and warm season Line 243 - of the cold and warm Line 321 - the cold and warm season Line 325 - the targetedAuthor Response
Please see the attachment.
